# Community-Centered Patient Journey Map in Opioid Use Disorder: A Tool to Address Compassion Fatigue among Community Pharmacists

**DOI:** 10.3390/pharmacy11020052

**Published:** 2023-03-14

**Authors:** Kenneth Charles Hohmeier, Alina Cernasev, Christina Leibold, Todd M. Moore, Erica Schlesinger, Gerald Cochran, Ileana Arce, Wesley Geminn, Marie Chisholm-Burns

**Affiliations:** 1Department of Clinical Pharmacy and Translational Science, College of Pharmacy, University of Tennessee Health Science Center, Nashville, TN 37211, USA; 2Department of Psychology, University of Tennessee, Knoxville, TN 37996, USA; 3Tennessee Department of Mental Health & Substance Abuse Services, Nashville, TN 37243, USA; 4Division of Epidemiology, University of Utah, Salt Lake City, UT 84112, USA; 5Office of the Provost, Oregon Health & Science University, Portland, OR 97239, USA

**Keywords:** pharmacist, addiction, medications for opioid use disorder, community pharmacy

## Abstract

Community pharmacists have become increasingly exposed to opioid use disorders in recent decades. However, both pharmacist training and traditional practice environments have not been adequate to prepare the pharmacist for both the patient care needs and regulatory barriers of patients experiencing opioid use disorders (OUD). As a result, there is a need to increase pharmacists’ awareness of both the overall patient experience as they navigate their OUD and the role of the community pharmacy as a touchpoint within that experience. To this end, a Community-Centered Patient Journey in Drug Addiction Treatment journey map was developed with expert insights, clinical experience, and in-depth interviews (conducted in spring of 2021) with 16 participants enrolled in licensed opioid treatment programs in Tennessee. Patients, policymakers, clinicians, and academic researchers were involved in the map development. Lived experiences of key informants were captured via in-depth interviews. A consensus decision-making approach was used throughout the patient journey map development process. The final patient journey map illustrates a non-linear pathway, describes the central role of the patient’s community, and emphasizes three major “pain points” within the system (access, adherence, and affordability). Future research should investigate the impact of such a journey map on pharmacy personnel’s knowledge, attitudes, and behaviors.

## 1. Introduction

Opioid use disorder (OUD) is widespread across the United States, with approximately two million Americans affected [1,2,3]. Consequently, opioid overdose deaths have continued to rise over the past three decades, with just over 80,000 deaths in 2021 [4]. One contributor to the growth in both OUD and opioid overdose deaths has been the increase in prescription opioids dispensed. From 1997 to 2007, there was an 866% increase in oxycodone sales and a 280% increase in hydrocodone sales in U.S. community pharmacies [5]. In 2020 alone, 142 million prescription opioids were dispensed, at a rate of 43.3 prescription opioids per person [6]. Approximately 14% of patients misuse prescription opioid medications, although this ranges depending on population between 5 and 25% [7,8,9]. This is concerning, as 4–6% of those who misuse prescription opioids move onto heroin, and approximately 80% of those who use heroin had previously misused prescription opioids [10,11,12]. Moreover, it is estimated that about one in four who use heroin develop OUD [13].

Community pharmacists have increasingly become exposed to OUD given these trends in opioid dispensing [14,15], yet their traditional role, experience, training, and education may ill-prepare them for the current practice demands of balancing OUD patient care and regulatory responsibilities. Moreover, given the role prescription opioids play in opioid use disorder, addiction, and overdose, the community pharmacy represents a critical touchpoint for positively impacting opioid-related morbidity and mortality. This is further emphasized by community pharmacies’ accessibility, with approximately 95% of Americans living within 5 miles of a community pharmacy [16]. For instance, if community pharmacies were used as an opioid treatment program site for pharmacy-based methadone dispensing, drive time to treatment would be reduced by as much as 75% [17]. For this reason, it has been suggested that community pharmacies should be further leveraged in the prevention, surveillance, and treatment of OUD [18]. Emerging evidence suggests that patients and physicians desire greater pharmacist involvement in OUD care. In a recent study investigating a collaborative care model between physicians and pharmacists in which buprenorphine care was managed by a community pharmacist, patients maintained high retention and adherence rates, with 90% of patients endorsing the model as very satisfactory [19]. In this latter example, the community pharmacist managed both follow-up buprenorphine care management and monthly buprenorphine dispensing responsibilities after initial screening, and assessment was performed by the overseeing physician [19].

Pharmacists also desire more responsibility in opioid-related care of their patients [20,21]. In the U.S., engaging pharmacists in treatment has been limited, as pharmacists have been placed in a position of regulatory enforcement of opioid use, while at the same time are professionally obligated to maintain a patient care advocate role [21]. Despite their professional obligation to patient care, pharmacists have been primarily charged by both the U.S. Drug Enforcement Agency and each state’s Board of Pharmacy to verify each opioid prescription is used for a legitimate reason and monitor for diversion, and to prove this routinely through stringent application of opioid-related regulations—leading to a clearly articulated regulatory role for pharmacists in delaying or denying opioid prescriptions [19]. However, pharmacists in the U.S. have indicated their desire to intervene to reduce high-risk opioid use behaviors, but complain that the lack of clear regulatory or clinical guidelines in the U.S. have relegated their opioid-related scope of practice solely to opioid regulation enforcement [20]. These opposing roles have led to reported “compassion fatigue” among frontline community pharmacy personnel in caring for these patients [21,22]. Pharmacists experiencing compassion fatigue may experience increasing frustration with patients, leading to negative emotional responses and decreased job satisfaction [23]. It is likely that the negative experiences with community pharmacists seen by some patients engaging in OUD treatment are due to this compassion fatigue [24]. For this reason, compassion fatigue may serve as a substantial, though latent, barrier to the community pharmacy’s engagement in care for patients with OUD.

One evidence-based solution to compassion fatigue is education [23]. Specifically, narrative messages that describe the patient’s experience and overall healthcare journey have been shown to reduce stigma and create a more holistic picture of the patient’s experience with OUD [25]. Narrative messages involve both emphasizing the external factors which play a role in the patient’s health status, and acknowledging personal responsibility in making healthy choices. Research suggests that a more comprehensive understanding of the patient’s experience “…can influence attitudes, behavioral intentions, causal attributions, and support for policy responses to health issues” [25,26]. Unlike physicians, nurses, and counselors who have the ability to screen and intervene on patients with OUD throughout their care continuum, community pharmacists experience a fragmented view of the patient’s journey, which centers primarily on pre-OUD diagnosis and treatment. Moreover, because of their physical separation from OUD treatment providers, inability to legally dispense certain medications for OUD, and regulatory responsibilities to police potential risky opioid use behaviors, community pharmacists are unlikely within their practice environment to fully experience this “comprehensive understanding of the patient’s experience” which is requisite to influence their attitudes, behaviors, and downstream experience of compassion fatigue.

Core to understanding the patient experience is connecting the patient’s journey in seeking and receiving care between interactions with the healthcare system—something referred to as “touchpoints” [27]. This can be an important link between what providers, such as pharmacists, “see” when they interact with the patient and the more complete view of the patient, including feelings, emotions, motivations, and attitudes as they live their lives with a chronic health condition [27]. A patient journey map is a formal exercise used to understand the patient’s journey with a specific healthcare condition from the patient’s perspective, typically for the purposes of improving the patient’s experience [28]. Patient journey mapping in OUD may allow stakeholders, such as pharmacists, to better understand the overall patient’s experience of the treatment cascade beyond the patient–provider interaction in a healthcare setting.

The objective of this project was to develop a patient journey map of OUD within the community pharmacy setting to serve as a future educational tool for pharmacists to better support patients with OUD.

## 2. Materials and Methods

### 2.1. Creation of the Patient Journey Map

A comprehensive approach to developing the patient journey map was employed using elements from Trebble et al., 2010, and McCarthy et al., 2016 [26,29]. In-depth key informant interviews with participants undergoing medication for opioid use disorder (MOUD) care were conducted across Tennessee. Interviews were audio-recorded, transcribed, and analyzed using thematic analysis. Thematic analysis includes the development of codes (e.g., words, phrases, or sentences) to represent patterns of meaning within qualitative data [30]. After analyzing the approximately 20 h of key informant interviews, all codes were extracted and further analyzed to develop the patient journey map.

To develop the map itself, we first created three separate visualizations: (1) a concentric grid with actors within a system, (2) a mind map of actors and their interdependencies (Figure 1), and (3) a perpendicular-axis grid representing the components of each phase of the pathway (Figure 2). These visualizations allowed the research team to see the patients’ journey from three distinct vantage points. Then, the team worked collaboratively to populate and review each visualization, and a consensus was reached on each element before proceeding to the next element.

The concentric grid was developed with the patient in the center, followed by their community, defined as the area where they live, work, and interact with their primary social system, and then finally, an outer area representing actors external to the patient’s community. Our aim was to define and visualize touchpoints and their proximity to the patient. Next, a mind map of those actors and their interdependencies was created to understand moderators and mediators within the system. Lastly, a perpendicular axis grid was created. Here, we developed a grid with an *x*-axis representing each of the traditional six phases of the patient journey, based on Trebble and McCarthy [26,29]: (1) Trigger Event/Awareness, (2) Help, (3) Care, (4) Treatment, (5) Behavioral/Lifestyle Change, and (6) Ongoing Care/Proactive Health. The *y*-axis included: (1) Touchpoints, (2) Moments of Truth, (3) Emotions, (4) Quotes, and (5) Pain Points.

The “Trigger Event” (or Awareness) phase during a patient’s journey is the point at which the patient becomes conscious of the need to seek medical help. During the subsequent stage, “Help,” the patient begins the process of identifying means to access treatment—either through peer or healthcare professional facilitation. Next, the patient receives “Care” through formal interaction with the healthcare system and healthcare workers, which is subsequently followed by “Treatment”, where a formal assessment and treatment plan are outlined by a medical professional. The “Behavior Change” phase follows since there is always a need for the patient to alter their lifestyle or behaviors beyond remaining adherent to prescription drug treatment. Finally, “Ongoing Care” involves the patient remaining in treatment chronically to manage their OUD.

At each phase of the patient’s journey, the patient has a range of experiences, emotions, and interactions which can be captured and mapped against each of the journey phases. A “Touchpoint” is the moment when the patient interacts directly with the healthcare system. “Moments of Truth” are gaps between the desired patient experience and the actual one. Before, during, and after interaction with the healthcare system, the patient will experience a range of “Emotions” related to their OUD, and these can often be captured simply by one or two words (e.g., concerned, mistrusting). Finally, what a patient states upon interview or survey can often “sum up” a journey phase with a few “Quotes.” Lastly, “Pain Points” are areas where patients encounter specific barriers or negative experiences related to their care.

### 2.2. Research Team

The patient journey mapping team included university researchers, state health officials, and clinicians. University researchers included faculty from the University of Tennessee Health Science Center and the University of Utah. State health officials included members of the Tennessee Department of Mental Health and Substance Abuse (TDMHSAS), including the Tennessee State Opioid Treatment Authority and Assistant Chief Pharmacist for TDMHSAS. Additionally, feedback was solicited from clinicians at the licensed opioid treatment programs. In total, there were five key members who aided in the creation of the map. AC is a PhD-trained qualitative researcher in social and behavioral pharmacy, KH is a community pharmacist and PharmD with training and experience in qualitative research, and IA is a PharmD in post-graduate training with specific training in qualitative research.

### 2.3. Participants and Data Collection

Based on a thorough review of the literature and conversations with Tennessee state officials with expertise in OUD, it was determined that there was a critical need to capture the voice of the patient population receiving methadone treatment within opioid treatment programs. Therefore, 16 patients who were enrolled in one of 15 Tennessee-based opioid treatment program clinics served as key informants for semi-structured interviews with university researchers. The Tennessee State Opioid Treatment Authority and Assistant Chief Pharmacist at TDMHSAS worked with licensed opioid treatment program facilities within the state to recruit patients. Qualitative research methods were used to perform telephonic interviews at opioid treatment program clinics (i.e., methadone clinics) across east, middle, and west Tennessee in the spring of 2021. A semi-structured interview guide was developed by the team and pilot-tested to simulate the intended key informants in the field. Interviews were recorded and transcribed confidentially by a third-party transcription service. Interviews were conducted by university researchers without any prior relationship with the participants (AC, KH, and IA). All transcribed interviews were deductively coded by the same pair of research team members (KH and AC) who conducted the interviews. Coding was based on a patient journey mapping approach and was analyzed using NVivo for Mac (QSR International; Burlington, MA, USA). Transcripts were not able to be returned to participants for review. It was determined a priori that recruitment would cease at a point of saturation, at which no new information was uncovered in subsequent interviews. The research was approved by the UTHSC Institutional Review Board (IRB). The final patient journey map was developed through a consensus decision-making approach including several rounds of virtual meetings, feedback, and revisions.

## 3. Results

### 3.1. Overview of Results

At the point of saturation, 18 interviews had been conducted across middle and eastern Tennessee. The average interview length was 65.6 min. Ten participants were male, and eight female. Participants were Black (2, 11%), White (14, 78%), or declined to answer (11%). Age ranges included 20–29 years (3, 17%), 30–39 years (3, 17%), 40–49 years (8, 50%), 50–59 years (2, 11%), and 60–69 years (2, 11%). Upon side-by-side comparison of the three patient journey map visualizations, the team decided the patient journey itself is best represented across several non-linear, interconnected roadways and across the three main settings of the map: the patient’s community, acute care, and chronic care (Figure 3).

### 3.2. Community-Centered Patient Journey Map

The Community-Centered Patient Journey Map in Drug Addiction Treatment places the patient’s community at the center of their journey. This “community centrality” in combination with the non-linear patient journey pathway presents an atypical patient care journey. A typical patient journey map is linear and flows in a logical phase-by-phase manner from awareness of disease through to treatment and chronic management [26,29]. In contrast, the Community-Centered Patient Journey, as articulated by participants and community stakeholders, is represented by an infinite loop of awareness of OUD, treatment and chronic management, the potential for relapse, and the opportunity to re-engage in treatment.

A constant variable within the patient’s journey was their community, represented by home—family, friends, and peers (i.e., social support structure)—and the culture of the environment around them. Each of these elements exerts influence on the patient, and their desire to seek or remain in treatment. For instance, negative peer pressure and a culture that promotes both low self-esteem and self-worth may work synergistically to push an individual into OUD. The size of the community pathway “loop” within the map represents both the large degree of time spent within it and its overall influence on the journey of addiction treatment. On either side of that journey are the patient’s experience with OUD and acute care (left-hand side of Figure 3) and chronic care (right-hand side of Figure 3). Within each of the community, acute care, and chronic care “loops” are salient variables captured on the perpendicular axis grid (Figure 2), including: touchpoints, moments of truth, emotions, supporting quotes, and pain points.

Throughout the patient journey map, highways are the best maintained and have the appearance of being intentionally designed by experts who oversee the system. Conversely, paths are unpaved, less direct, and not formally considered part of the highway system but are often used as unideal but necessary alternatives to the highways system by the patient. Scattered throughout both the highways and paths are obstacles, or “pain points”. These “pain points” centered across three main sub-themes: accessibility, adherence, and affordability (Figure 3, cloud text in the sky). Roadway signage is informative, but numerous and distracting, representing the lack of clear direction or pathways to the care available for patients with OUD. There are also great distances between healthcare touchpoints, where the patient and the healthcare system interact.

### 3.3. Beginning the Patient Journey toward Treatment

The journey stars within the patient’s community, where there is an exposure to an opioid (either licit or illicit), with or without peer pressure. Red arrows are used to demonstrate the flow into addiction either from opioid access from peers (illicit) or from a valid prescription (licit). OUD can continue within the community indefinitely until what participants referred to as a “trigger” event (i.e., a critical moment where an individual is made acutely aware of the negative consequences of their opioid use and the need for behavior change) or an “ah-ha” moment of general awareness of the need to seek help (represented by the green boxes labeled “Exit 1A” and “Exit 1B”). These different events can lead to two different pathways. The ‘general awareness’ pathway (“Exit 1A”) does not include an acute negative event, but rather a personal choice to proceed to chronic care and treatment. In contrast, a ‘trigger event’ (“Exit 1 B”) is another pathway to treatment, but requires an acute, negative event such as overdose, incarceration, or hospital or rehabilitation center admission. In this latter example, the pathway begins with a dirt road, rather than a paved one, to represent the distance the individual is from general society norms and the more severely difficult the path is toward treatment.

### 3.4. Entering Treatment through ‘Trigger Events’

Within the ‘Trigger events’ pathway, participants described several types, including: incarceration, drug court, arrest, opioid-related trauma (i.e., death within social circle due to OUD), or overdose. This pathway terminates with acute care being provided in some manner to address the immediate needs of the individual, including inpatient detoxification. However, the terminal point includes a formal re-entry back into the individual’s community. This is important, as participants noted that the community represented both the original genesis of their OUD, including access to both licit and illicit opioids and negative peer pressures, as well as supportive structures required for recovery (e.g., positive social structure, work, positive community activities).

“Yeah, you got to want to be clean, you know what I’m saying? Then, I didn’t care, you know what I mean? I was just using it for a crutch. But now I’m- I don’t want to go to prison, you know what I’m saying? I just, I want to do right, and that’s the difference. And like, my Dad, because he died of drug use, you know what I mean? He just died this past year.”(P12)

### 3.5. Entering Treatment through Awareness

Participants described a separate, chronic care pathway for outpatient OUD treatment, termed “Awareness” (“Exit 1A”). This pathway was used by participants either when they had a ‘general awareness’ of the need for treatment (e.g., distress, loneliness, sadness) or when they transitioned from acute care. This pathway is represented by a completely paved road to indicate that this entry into care is more formalized, more closely connected to societal norms, and in closer proximity to the healthcare system. However, the majority of the pathway is still located outside of the patient’s community, representing a potential barrier to access.

“Well, I needed somebody that would kind of, how do I put this, you know [help me]. Because, when I was going to the doctor to get the pain pills, I’d get 120, 90 to 120 [pills], and that’s a 30-day supply. Well, I would be out in two weeks. That’s a lot of pills. So I said, something has got to stop. I’m not going to survive doing this. And I was addicted. They would say, well, why don’t you just quit? Well, I wish it was easier said than done! I just, it’s a lot of stuff that plays into it besides just being an addict and addicted to pain pills. You’ve got all the peer pressure and you have to change your life. You can’t be around people that associate with you. You know, you lose friends. It’s just the way it is if you want to quit, you know, like that. But I got so many [painful] injuries- like, right now, I’m going to have to eventually have a whole hip replacement in my right hip now. And I just, you know, just keep holding on, I don’t want to have to go do it. But I’m eventually going to have to do it. It’s just I’m at that age that I’m falling apart.”(P9)

“I’m 39, and I’ve been- since I was 25, I’d been in and out of drugs, not hard drugs, it was mostly just pot, and then I started taking Lortabs, and then it went to OxyContin, which then led me here to methadone. So I guess my goal is to just get clean…”(P2)

The “Awareness & Ongoing Care” pathway (“Exit 1A”) was described by participants as including several barriers: treatment stigma, cost of care, wait lists for treatment, and travel distances to treatment.

“Once you get two days behind [in payment], you’re put on financial detox. They take you down ten milligrams every day until you’re down to zero, and then you’re done. They just kick you out.” (P2)

Those participants who had previously received chronic care for OUD remarked that these barriers had directly led to past instances of treatment non-adherence or relapse. This is visually represented by a dirt “trail”, which is an undesirable exit from treatment back into their community, and potential for relapse. For methadone treatment in particular, participants noted that the requirement to visit the clinic most or all days each week to receive their dose of methadone would interfere with keeping their job or restrict which jobs they could work.

“That was frustrating because, when I was working a day job, there was a time or two that I had to not take my medicine because I had to be at work by 9:00. And, I mean, I would get there [to the clinic] at 6:30 or 6:45, and I would have to leave at like 8:30. I’m like, I can’t wait. I’m like, you know, you guys want me to stay clean, believe you me, I want to stay clean, too, but I got to have a job too.”(P11)

The scarcity of treatment access points directly created the barriers of treatment “wait lists” and travel distance barriers. Some participants remarked that travel to clinics would take over an hour, one way.

“And, at that time, they had like three to six months waiting list, and I thought, oh my God, I can’t wait that long.”(P15)

“…I have to drive an hour every day. I’ve lost like some good jobs because I can’t never work first shift.”(P8)

Throughout treatment, patients described the treatment-related stigma that was ever-present.

“Yes, you do [feel stigmatized]. You know, and it may be self-thought. You know, you may have got the wrong attitude from the pharmacist or the personnel there at the pharmacy or whatever. It just seems like sometimes, you know, if you’re picking up a medication, sometimes you’re treated differently than- I’m treated differently when I go in to pick up my heart medication, okay? If I go in and get my blood pressure medication or my blood thinners, I’ve had several heart attacks, so if I go in and get that medication, I’m smiled at and sent on my way. But sometimes when I went in to get the pain medication, [medication for OUD], you know, you could just feel a sense that you were treated a little differently. And I try not to think that about anyone, I really do, but sometimes it’s hard not to feel that. You know, you sense that you’re being treated differently, yeah.”

### 3.6. Returning to the Community and Ongoing Care

After exiting either “Awareness & Ongoing Care” pathway (“Exit 1A”) or a ‘trigger event’ (“Exit 1 B”), the patient returns to their community to continue their OUD treatment. As described earlier, this is also where the patient is initially exposed to either licit or illicit opioids. Therefore, the patient’s entire opioid experience surrounds the community surrounding where they live and work.

As pictured in Figure 3, medical professionals were primarily described as being located outside of the community (e.g., hospitals, medical offices, opioid treatment programs). Their influence on the patient while within the community is consequently limited to physician–patient touchpoints, and this was articulated by individuals. The exception to this was the community pharmacist, a medical professional located within the patient’s community and with whom individuals would frequently interact even between medical office visits. However, despite their accessibility, the perceptions of the community pharmacist and their role in care represented a “pain point,” whereby the experience preferred by the patient with the pharmacist was not always what was experienced.

“You know, you may have got the wrong attitude from the pharmacist or the personnel there at the pharmacy or whatever. It just seems like sometimes, you know, if you’re picking up a narcotic medication, sometimes you’re treated differently than- I’m treated differently when I go in to pick up my heart medication, okay? If I go in and get my blood pressure medication or my blood thinners, I’ve had several heart attacks, so if I go in and get that medication, I’m smiled at and sent on my way. But sometimes when I went in to get the pain medication, narcotics, you know, you could just feel a sense that you were treated a little differently. And I try not to think that about anyone, I really do, but sometimes it’s hard not to feel that. You know, you sense that you’re being treated differently.”(P14)

One patient detailed their positive experience with a community pharmacy in another country, which allowed the community pharmacist to be involved in OUD treatment with methadone to improve treatment access.

“And, then the biggest difference was [the prescriber] gave me a written prescription and just told me to go to a pharmacy and get it filled. They don’t really know me, I’m just the person who shows up, but of course [the prescriber and community pharmacist] have talked. That’s how they do it, the [prescriber and community pharmacist] talk to each other and set it up between them, so they know you’re coming, right?”(P16)

Other positive moderators and mediators found within the community “loop”, which were noted to encourage overall recovery and ongoing care, visualized as the “fence” in the middle of the community, hidden by trees and only accessible by a “hidden drive.” This represents what participants articulated as difficult-to-find resources and sources of support which could be found within the same community where their OUD began. This included meaningful employment, positive social connections (e.g., family, friends, partners), and clinicians (e.g., counselors, physicians, pharmacists) who supported their recovery.

“Oh, well, I was able to keep a job. My social life improved. I wasn’t always looking or trying to find, you know, my fix just so I could feel better and go sit in my room. Once I got the treatment, I was able to get back out there. And I’m a social butterfly, so I like being around people and talking and hanging out with friends and doing stuff. So once I got treatment, it kind of opened my life back up.”(P6)

## 4. Discussion

This paper presents a novel patient journey map for those engaging in substance use disorder treatment. Notably, the data support a patient journey line that is non-linear, which is a substantial contrast to typical patient journey maps with a clear beginning and end. This visualization captures the sometimes-revolving cycle of treatment, relapse, and recovery common in OUD. The presented journey map also places the community at the center, as the patient’s experience revolves around their community and is largely influenced by factors within the community, including “touchpoints” with the community pharmacy. Moreover, this community-centered lens is important because of the biopsychosocial etiology of OUD, and its necessary connection to the home environment, social connections, and other community-related factors [31]. The result is a visual map which provides a more comprehensive view of where a patient may be at the time of presentation to a healthcare provider, especially the community pharmacist.

It is of particular note that in the center of this Community-Centered Patient Journey Map in Drug Addiction Treatment, discussed in Figure 3, are community resources. These include peers, the overall culture of the environment, and the community pharmacy. Counselors, other clinicians, and medical facilities—while all available in the outpatient setting—are not typically available in the patient’s own community, as articulated by the participants in this study and also seen in the supporting literature [17]. The centrality of the community pharmacy within the journey map makes visible what has been historically invisible to stakeholders of OUD treatment—the underutilization of the pharmacist. Moreover, it also makes visible to pharmacists the broader care journey the patient undergoes, and may help as a training, educational, and patient care resource for pharmacists to understand how best to care for these individuals.

This depiction of the community pharmacy within the continuum of a patient’s experience with OUD is the first, to the authors’ knowledge. Community pharmacists have historically been either excluded from OUD care and treatment guidelines or relegated to roles related to regulatory enforcement (e.g., accessing prescription-monitoring databases, denying or postponing medication fills) [20,32]. This has placed the community pharmacist in a difficult position where they are unable to intervene in risky opioid use or OUD to improve patient care, but are able to restrict opioid prescription access as required by federal and state regulations. As a consequence, the patient–pharmacist relationship has suffered, with pharmacists reporting compassion fatigue and patients reporting negative experiences at the community pharmacy [21,22].

Compassion fatigue in opioid use disorder is seen across professional settings, including pharmacy, nursing, medicine, and emergency service personnel [21,22,23,33,34]. Comprehensive models of compassion fatigue show that several external factors predispose healthcare professionals to compassion fatigue, including lack of positive patient outcomes, negative patient interactions, and lack of resources to adequately address patient needs [35]. Prior research indicates that each of these elements is present in the community pharmacy setting, as personnel indicate that the lack of clear professional guidelines, regulatory requirements, and scope of practice limitations do not allow the pharmacist to provide care for patients [20,21].

It has been suggested that pharmacist education may improve pharmacist-provided OUD care [7,14,15,32]; however, such education that excludes a fundamental discussion about the role and overall patient experience may be inadequate. Although not a holistic solution, providing frontline community pharmacists with training on opioid misuse, use disorder, and addiction may serve to offset compassion fatigue in this patient population. In general, health provider attitudes toward OUD and OUD treatment have been positively impacted by formal training programs [36,37,38]. This has also been found to be true for pharmacists’ attitudes toward OUD [39]. Notably, pharmacists’ attitudes toward patients with OUD improved with increasing exposure to patients filling prescriptions for OUD treatment, seeing patients improve over the course of OUD treatment, and seeing the positive impact a pharmacist can have on the course of OUD care [40]. In short, as the pharmacist realized the “big picture” of OUD care, they were better able to process the negative experiences that occurred with patients misusing opioid prescriptions. It is the authors’ feeling that the present journey map may be a helpful resource in training and educating pharmacists on this “big picture,” although future prospective studies will be required to understand if and to what degree this is true.

There were several limitations to this study. Data used to derive the patient journey map were gathered from a single state in the southeastern United States. Moreover, data relied solely on qualitative data and expert opinion. Although this data is sufficiently rigorous to develop patient journey maps [26,28], future research should incorporate a mixed-methods approach across geographic regions. Finally, the patient journey map has not been validated for its use in compassion fatigue with community pharmacists. Further research is needed to incorporate the journey map into an educational intervention to identify its impact on pharmacists’ attitude and behaviors, as well as patient care outcomes and patient care satisfaction.

## 5. Conclusions

This patient journey map depicts the patient’s perspective in managing OUD across a continuum of care settings. The patient journey map was developed to serve as a potential educational tool for pharmacists to better support patients with OUD. A non-linear patient journey is depicted when navigating OUD treatment, thus the map depicts a circular trajectory including both treatment and relapse. The map also indicates that the pharmacist may stand to play a critical role in both medication treatment access and in the identification and referral of patients into treatment. Future research may be useful to validate and build from this understanding so that care for patients with OUD in community pharmacies can be enhanced.

## Figures and Tables

**Figure 1 pharmacy-11-00052-f001:**
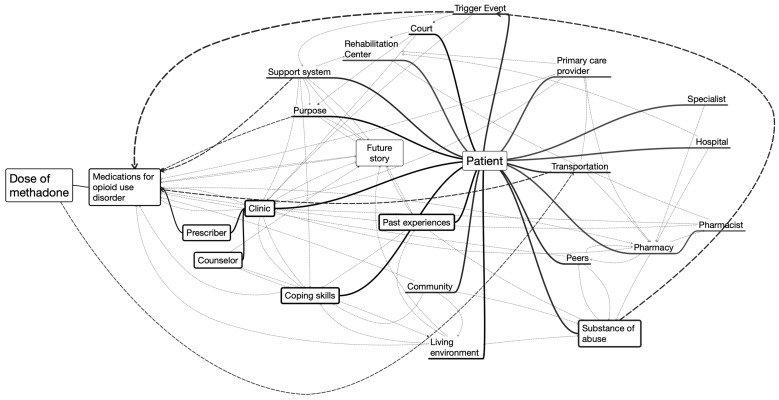
Mind Map of Actors and Their Interdependencies of the Patient Journey in Opioid Use Disorder Treatment.

**Figure 2 pharmacy-11-00052-f002:**
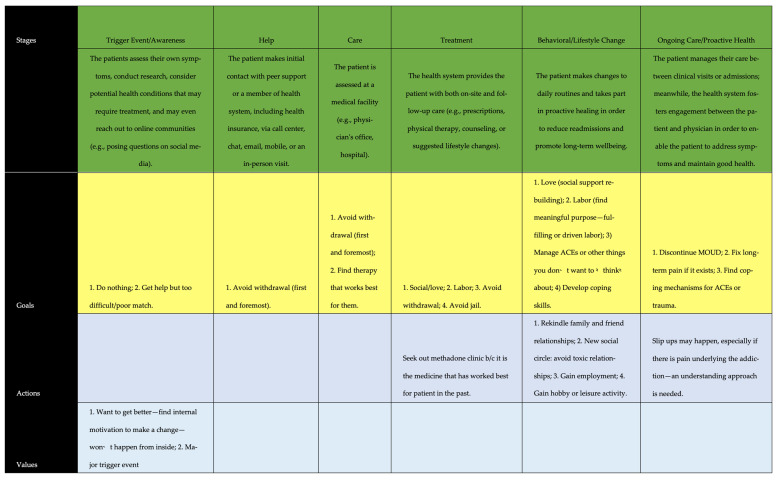
Sample Draft of the Perpendicular Axis Grid of the Patient Journey in Opioid Use Disorder Treatme. ACE = Adverse childhood experiences; ED = Emergency department; MOUD = Medication for opioid use disorder.

**Figure 3 pharmacy-11-00052-f003:**
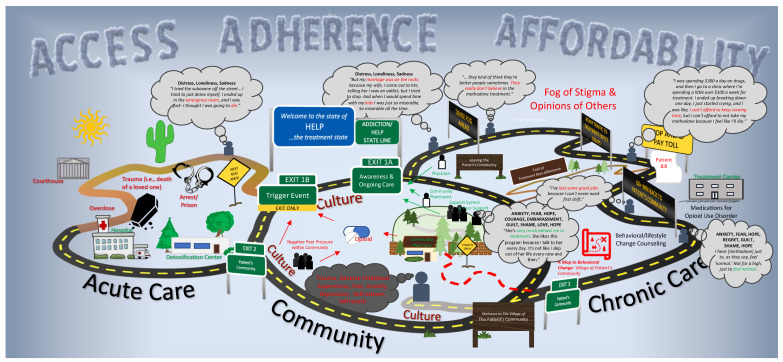
Community-Centered Patient Journey Map in Drug Addiction Treatment.

## Data Availability

Not applicable.

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
