# Peer review of "Community-Centered Patient Journey Map in Opioid Use Disorder: A Tool to Address Compassion Fatigue among Community Pharmacists"

_pharmacy, 2023, doi:10.3390/pharmacy11020052_

Round 1

Reviewer 1 Report

Title: “Community-centered Patient Journey in Opioid Use Disorder 2 Treatment and Medication Adherence” – Where?

Abstract

- “regulatory needs of patients”; please give more details.

- When was the study carried out? Timeframe?

- “map development”; please give more details

Keywords: please use some MeSH terms; preferably, keywords should not be used in tittle and abstract

-          Please, explain all concepts when first introduced in the paper. New introduced concepts should be explained. Please check all the paper.

Introduction

-          “In a recent study investigating a collaborative care model between physicians 57 and pharmacists in which buprenorphine care was managed by a community pharmacist”; please give more details about this program

-          “However, engaging pharmacists in treatment has been limited as pharmacists  have been placed in a position of regulatory enforcement of opioid use; as well as a patient care advocate role”; Why? This is not clear. Please note that the concept of “regulatory enforcement” is variable between different countries. Please explain US regulation, regarding opioid use. The present paper will be read by an international audience.

-          “This, in turn, has led to reported “compassion fatigue””; Why? Patients not follow the advice of community pharmacists?

-          Please give more details/examples about concrete interventions of community pharmacists.

-          Please cite some systematic reviews and/or meta-analysis on the present topic (or related topics). These works should also be cited in discussion.

2. Materials and Methods

- Avoid the use of too many abbreviations.

- What Equator guidelines have been followed?

- Please see: https://www.equator-network.org/?post_type=eq_guidelines&eq_guidelines_study_design=qualitative-research&eq_guidelines_clinical_specialty=0&eq_guidelines_report_section=0&s=

- Allison Tong, Peter Sainsbury, Jonathan Craig, Consolidated criteria for reporting qualitative research (COREQ): a 32-item checklist for interviews and focus groups, International Journal for Quality in Health Care, Volume 19, Issue 6, December 2007, Pages 349–357, https://doi.org/10.1093/intqhc/mzm042

- Please briefly define/explain between brackets (or in a Table) points 1 to 5:  (1) Trigger Event/Awareness, (2) Help, (3) Care, (4) Treatment, (5) Behavioral/Lifestyle Change, and (6) Ongoing Care/Pro-active Health. The y-axis included: (1) Touchpoints, (2) Moments of Truth, (3) Emotions, 118 (4) Quotes, and (5) Pain Points (lines 116 to 119)

- Lines 103-106: “To develop the map itself, we first created three separate visualizations: 1) concentric grid with actors within a system, 2) mind map of actors and their interdependencies (Figure 1), and 3) perpendicular axis grid representing the components of each phase of the  pathway (Figure 2).” Please give more details in the text, regarding points 1 to 3. This is not easy to understand without consulting Figures.

- Please try to improve Figures readability. Figures/Tables are not very clear.

- Please check the format of figures and Tables in instructions to authors

2.2. Research Team

- What experience or training did the researcher have? Demographic data? Training?

- Relationship with participants: Was a relationship established prior to study commencement? What did the participants know about the researcher? What characteristics were reported about the interviewer/facilitator?

Please see: Consolidated criteria for reporting qualitative studies (COREQ): 32-item checklist; and check if all items are covered

2.3. Participants and Data Collection

- NVivo for Mac. Reference?

- How many people refused to participate or dropped out? Reasons?

- What was the duration of the interviews?

- Were transcripts returned to participants for comment and/or correction?

- Did participants provide feedback on the findings?

- Please see: Consolidated criteria for reporting qualitative studies (COREQ): 32-item checklist; and check if all items are covered

- Figure 3: please present the full meaning of abbreviations below all figures.

3. Results

- “Upon side-by-side comparison of the three visualizations, the team decided the patient journey itself is best represented across several non-linear, interconnected roadways  and across the three main settings of the map: the patient’s community, acute care, and chronic care (Figure 3).” Why? How? Please present this issue in methods.

- Please consider the creation of subheadings in Results. Subheadings are very useful and exponentially increase text readability/comprehension.

Discussion

-          New references cited in introduction, should be here recited.

-          Please discuss potential study biases. For instance, see Balhara KS, Weygandt PL, Ehmann MR, Regan L. Navigating Bias on Interview Day: Strategies for Charting an Inclusive and Equitable Course. J Grad Med Educ. 2021 Aug;13(4):466-470. doi: 10.4300/JGME-D-21-00001.1. Epub 2021 Aug 13. PMID: 34434507; PMCID: PMC8370377.

-          Importantly, please discuss/suggest possible/concrete community pharmacists’ interventions. Please give some examples. Have the community pharmacies from USA real programs on the management of opioids crisis? How can they be improved?

-          Please create a section about practical implications, future research, and study limitations at the end of introduction. Please use subheadings to group these topics.

Author Response

Thank you for your time and insights. Please find attached our response.

Reviewer 2 Report

1. I would recommend reconsidering the title to include the word "map" and what the purpose of the creation was. For example it could say "Community-centered Patient Journey Map in Opioid Use Disorder: A potential tool to combat compassion fatigue among community pharmacists" 

2. In lines 39-40 what is the difference in population for the wide range from 5-25%?

3. The link between the data presented for opioid misuse and the significant of it evolving to heroin use is unclear in the introduction's first paragraph. 

4. Comma is needed between role and experience in line 45.

5. Could you expand on how pharmacists desire more responsibility in opioid-related care? (line 61)

6. It may be helpful to have more information about the patient demographics who participated in the interview process (ie patient ages, sex, years in treatment, number of relapses).

7. For Figure 1 I would recommend making the font a larger size. 

8. For Figure 2 I would recommend ensuring each column is clearly delineated with lines. 

9. For Figure 3 I would recommend making this bigger. 

10. There is a typo in line 144 ("a-priori")

11. I would consider re-writing the discussion to include more data to support the use of patient journey maps as an educational tool to combat compassion fatigue. 

Author Response

(The authors gave the same response as above.)

Round 2

Reviewer 1 Report

Dear Authors, Thanks for all updates. This paper is a good piece of scientific research and information. The paper is innovative and disruptive, regarding methodologies. Congratulations.

Minor comments:

-         - Please describe and discuss Figure 3 in Discussion.

Author Response

Thank you for the feedback and positive notes. We've added the additional paragraph discussing Figure 3.